# Cross-Reactivity Conferred by Homologous and Heterologous Prime-Boost A/H5 Influenza Vaccination Strategies in Humans: A Literature Review

**DOI:** 10.3390/vaccines9121465

**Published:** 2021-12-10

**Authors:** Adinda Kok, Ron A. M. Fouchier, Mathilde Richard

**Affiliations:** Department of Viroscience, Erasmus Medical Center, 3015 GD Rotterdam, The Netherlands; a.kok.2@erasmusmc.nl (A.K.); r.fouchier@erasmusmc.nl (R.A.M.F.)

**Keywords:** avian influenza A virus, H5 influenza virus, zoonosis, pandemic, vaccination

## Abstract

Avian influenza viruses from the A/H5 A/goose/Guangdong/1/1996 (GsGd) lineage pose a continuing threat to animal and human health. Since their emergence in 1997, these viruses have spread across multiple continents and have become enzootic in poultry. Additionally, over 800 cases of human infection with A/H5 GsGd viruses have been reported to date, which raises concerns about the potential for a new influenza virus pandemic. The continuous circulation of A/H5 GsGd viruses for over 20 years has resulted in the genetic and antigenic diversification of their hemagglutinin (HA) surface glycoprotein, which poses a serious challenge to pandemic preparedness and vaccine design. In the present article, clinical studies on A/H5 influenza vaccination strategies were reviewed to evaluate the breadth of antibody responses induced upon homologous and heterologous prime-boost vaccination strategies. Clinical data on immunological endpoints were extracted from studies and compiled into a dataset, which was used for the visualization and analysis of the height and breadth of humoral immune responses. Several aspects leading to high immunogenicity and/or cross-reactivity were identified, although the analysis was limited by the heterogeneity in study design and vaccine type used in the included studies. Consequently, crucial questions remain to be addressed in future studies on A/H5 GsGd vaccination strategies.

## 1. Introduction

Influenza A viruses pose a significant global burden on human and animal health. They are divided into subtypes based on the antigenic properties of their surface glycoproteins, the hemagglutinin (HA) and neuraminidase (NA). So far, 18 subtypes of HA and 9 subtypes of NA have been identified. Influenza A viruses infect a diverse range of species beyond their original reservoir—wild waterfowl—such as poultry, pigs, horses and humans. The broad host range of influenza A viruses creates opportunities for frequent interspecies transmission favouring zoonosis and pandemics. Influenza A virus zoonosis occurs when a virus from the animal reservoir crosses the species barrier and infects humans. Additionally, a pandemic can be sparked if such a zoonotic virus can also transmit efficiently in the—mostly immunologically naïve—population. These major threats to human health may have devastating socio-economic consequences, as observed currently with the COVID-19 pandemic, caused by a member of the coronavirus family.

Since 1918, four influenza A virus pandemics have occurred [1]. To date, only viruses carrying HAs of the H1, H2 and H3 subtypes and NAs of the N1 or N2 subtypes are known to have caused influenza A virus pandemics. Influenza A viruses of other HA and NA subtypes have caused numerous zoonotic infections [2]. In 1997, highly pathogenic avian influenza A/H5 viruses of the A/goose/Guangdong/1/1996 (GsGd) lineage were first detected in Hong Kong, resulting in the first reported zoonosis of influenza A viruses carrying an H5 HA [3]. Since 2004, viruses of the A/H5 GsGd lineage have spread to multiple countries over four continents and are still enzootic in poultry and wild birds in various parts of the world. As of June 2021, 862 cases of human infection with A/H5 GsGd viruses have been reported in 17 countries, of which 455 were fatal [4]. Although no sustained human-to-human transmission has been reported, it has been shown that only a handful of amino acid changes are sufficient for viruses from the A/H5 GsGd lineage to become transmissible via the air between mammals [5,6,7], which raises further concerns about a potential A/H5 influenza pandemic.

The continuous circulation of A/H5 GsGd viruses for over 20 years in many different species has resulted in the diversification of the H5 HA gene in numerous genetic clades and subclades [8,9]. Moreover, the HA proteins of the A/H5 GsGd viruses have diversified at the antigenic level, leading the WHO GISRS network to release a biannual update on A/H5 GsGd antigenic evolution and on the selection of candidate vaccine viruses for potential use in humans [10]. Viruses from different genetic and antigenic (sub)clades circulate concurrently, increasing the challenges linked to pandemic preparedness against A/H5 GsGd viruses. In case of a novel influenza virus pandemic, manufacturing a vaccine based on the pandemic strain is estimated to take 3 to 6 months, by which time the first infection wave may have passed in many countries [11]. Stockpiling pre-pandemic vaccines could offer at least a temporary solution to this problem, but only if these vaccines are immunogenic and induce antibodies which cross-react with the pandemic virus. Viruses from the A/H5 GsGd lineage pose an additional challenge since inactivated A/H5 influenza virus vaccines generally have low immunogenicity in humans as compared to inactivated seasonal influenza virus vaccines [12]. 

Several pre-pandemic vaccines have been licensed for A/H5 influenza viruses [13]. However, most of them are inactivated vaccines and based on old antigenic variants which thus might result in limited immunogenicity and cross-reactivity against currently circulating A/H5 GsGd viruses. To address the need for A/H5 influenza vaccines with increased immunogenicity and cross-reactivity, numerous clinical studies have been performed assessing the effect of adjuvant use, amount of HA antigen and different vaccine platforms, amongst others. Additionally, A/H5 influenza prime-boost vaccination strategies have been assessed in which two antigenically different vaccine strains are used in subsequent vaccinations. Such a heterologous prime-boost vaccination strategy could result in immune responses of increased breadth by boosting immune responses against common epitopes that might be shared with other antigenically distinct strains [14] and is thus of particular interest in the context of A/H5 pandemic preparedness. 

Here, we review the literature available on the breadth of antibody responses induced by A/H5 influenza virus vaccines in humans upon homologous and heterologous prime-boost vaccinations to identify aspects correlating with high immunogenicity and cross-reactivity, as measured by several immunological endpoints against the vaccine antigen and heterologous antigens.

## 2. Materials and Methods

The WHO overview of clinical evaluation of pandemic or potentially pandemic influenza vaccines [15] was used as a starting point for the selection of relevant scientific articles (Figure 1). This table was composed of 684 entries, of which 183 entries contained data on H5 vaccines. A total of 49 of these 183 entries were excluded because the results of the clinical trial had not been published. Preclinical studies in animals (*n* = 2), duplicate entries (*n* = 8) and articles in which no original research was described (*n* = 5) were also excluded. Given the fact that the scope of this review was to assess the breadth of the immune response upon vaccination, studies in which no humoral immunogenicity data (*n* = 10) or only humoral immunogenicity data to a single strain were reported (*n* = 51) were excluded. Subsequently, articles solely focused on pediatric or elderly populations (*n* = 6) were removed. Finally, articles from which no numerical values of immunological endpoints could be extracted were excluded (*n* = 14). From 39 of the remaining articles, the available data (reported in tabular format or as numerical values in graphs or text) on humoral immunological endpoints were extracted. These articles were divided in two groups, based on whether they described a homologous (*n* = 25) or heterologous (*n* = 14) prime-boost vaccination approach, as summarized in Table 1 and Table 2, respectively. Several articles describing a heterologous prime-boost regimen also contained data of homologous prime-boost regimens (highlighted in Table 2), which were included in the homologous dataset. 

Raw data on humoral immunological endpoints generated using three commonly used serological assays [55], namely hemagglutination inhibition (HI), single-radial hemolysis (SRH) and microneutralization (MN) assays, were extracted (Appendix A). The HI and SHR assays rely on the ability of influenza viruses to bind to erythrocytes. In the HI assay, antibodies which specifically inhibit agglutination of erythrocytes are detected and the antibody titer corresponds to the reciprocal of the highest serum dilution that inhibits agglutination. In the SRH assay, specific antibodies are quantified by measuring areas of hemolysis mediated by complement and induced by antigen-antibody complexes. In the MN assay, the extent to which antibodies inhibit virus entry and replication in cell culture systems is assessed. Since the SRH assay was used significantly less often than the HI and MN assays in the included studies and all studies in which SRH data were reported also included data obtained using HI and/or MN assays for the same set of serum samples, SRH data were excluded to narrow down the dataset for the present analysis while maximizing comparative analysis.

The data of these assays can be used to report several different immunological endpoints. Most simply, the geometric mean titer (GMT) observed in an experimental group can be reported. Moreover, the geometric mean ratio, also termed the geometric mean increase, can be reported, which is defined as the relative change in titer from pre- to post-vaccination. Since a geometric mean ratio is less relevant in a largely immunologically naïve population, which is the case for the participants in the present studies, this immunological endpoint was not included in the present analysis. Another reported outcome parameter is the percentage of subjects who would be seroprotected (SP), i.e., with an immune response equal to or above a predefined threshold. The established threshold for the HI assay is a titer of ≥1:40, which is considered protective at 50% against infection with human influenza viruses [56]. For MN assays, there is no standardized threshold for seroprotection, and in the present dataset, it varied from >1:20 to >1:40. Lastly, the percentage of subjects who seroconverted (SC) can be reported. Seroconversion refers to a predefined increase in titer upon vaccination or a titer above a certain threshold in absence of a detectable pre-vaccination titer. In all studies of the present dataset, a fourfold increase in titer for seroconversion was used for HI and MN assays in case of a pre-vaccination titer. For HI assay data, a threshold of ≥1:40 was used in all studies in case of a pre-vaccination titer below the detection limit, with the exception of one article in which a threshold of ≥1:20 was used [39]. For MN assay data, the detection limit of the assay and the threshold used for SC varied between <1:20 and <1:28 to ≥1:40 and ≥1:56, respectively. SC data were excluded when pre-booster titers were used as baseline, as opposed to pre-vaccination titers. Seventy percent of the data entries were comprised of serum samples obtained <100 days after the last vaccine dose administration, which also included at least one time point of each study. The remaining thirty percent of the data entries were comprised of serum samples obtained from 100 days to 3.5 years after the last vaccine administration. To allow for a better comparison between data from different studies, data obtained with serum samples collected >100 days after the last vaccine dose administration, were excluded from the main analysis but included in Appendix A. For the vast majority of the experimental groups of each study, raw data on at least one immunological endpoint were available and extracted for our dataset. In the few instances wherein no numerical raw data were available, the main outcome was discussed separately in this review.

## 3. Results

Compiling the data from the selected articles resulted in 231 data entries. Each entry corresponds to a study’s experimental group at a particular time point. For one entry, several immunological endpoints against the homologous and one or multiple heterologous antigens were included if reported in the corresponding article. 

### 3.1. Homologous Prime-Boost Vaccination Regimens

#### 3.1.1. Description of the Dataset

Several different vaccine types and adjuvants were represented in the included studies on homologous vaccination strategies (Table 1, Figure 2A). In these studies, data entries in which split inactivated vaccines were assessed were the most numerous (*n* = 87), representing about half of the whole dataset, followed by those from studies on subunit (*n* = 39), whole inactivated (*n* = 19) and virus-like particle (VLP) vaccines (*n* = 3). In the present dataset, split inactivated vaccines were combined with the oil-in-water adjuvants AS03A, AS03B (squalene, α-tocopherol, polysorbate 80) or AF03 (squalene, polyoxyethylene cetosteraryl ether, mannitol, sorbitan oleate), subunit vaccines with aluminum salt adjuvant or oil-in-water adjuvant MF59 (squalene, polysorbate 80, sorbitan trioleate) and whole inactivated and VLP vaccines with aluminum salt adjuvant. All these vaccine types except the VLP vaccine were also assessed without adjuvant. The amount of HA per dose varied from 1.9 to 90 μg in these vaccines, with 3.75 μg HA being the most frequently used. The homologous vaccination dataset also included experimental groups in which live attenuated (*n* = 7) and vector-based vaccines (*n* = 10) were assessed, the latter with either a modified vaccinia virus Ankara (MVA) vector (*n* = 8) or an adenovirus subtype 4 vector (*n* = 2) in combination with a subunit vaccine booster dose. 

The effect of multiple vaccine doses was assessed in the majority of the studies. A third of the present dataset is comprised of data obtained after one vaccine dose and half of the dataset is comprised of data obtained after two vaccine doses. On a few occasions, data were obtained after the administration of three or four doses. The time between the administration of the vaccine doses varied from seven days to a year. A three-week interval was most frequently used. Seven different vaccine antigens have been used, all of them early GsGd antigens. The majority of the data were generated using either A/Vietnam/1194/2004 (clade 1) or A/Indonesia/05/2005 (clade 2.1) as vaccine antigens (Figure 2A).

#### 3.1.2. Discussion of the Data

The height and breadth of antibody responses induced by homologous prime-boost vaccination strategies with A/H5 influenza vaccines were visualized by plotting the immunological endpoint values per entry obtained for the antigen homologous to the vaccine (x axis) against that obtained for a heterologous antigen (y axis) (Figure 3). Interactive visualizations of these data are shown in Appendix A, in which information about the vaccine type, adjuvant, dose of HA, timing between the vaccination and the blood draw and the antigen used in the vaccine are displayed. 

For all immunological endpoints, a positive trend was observed between the value obtained against the homologous antigen to the vaccine and that obtained against heterologous antigens. On some occasions, the immunological endpoint values against the homologous antigen were higher than that against the heterologous antigen (bottom right part of the graphs), but the reciprocal was not observed. It was evident from these data that the vast majority of the immunological endpoint values located in the bottom left part of the graphs (seroprotection and seroconversion rates below 50 % and log2 GMT below 6.5 for both homologous and heterologous antigens) were generated either upon one single vaccine dose (smallest size circles), upon multiple doses of non-adjuvanted inactivated vaccines (open circles) or upon vaccination with two doses of MVA or adjuvanted VLP vaccines. 

In all the studies in which the impact of a single versus multiple vaccine doses was assessed, immunological outcomes against both the vaccine and heterologous antigens were improved after two doses as compared to a single dose. After two doses of an adjuvanted vaccine (regardless of the type of vaccine, with the exception of the VLP vaccine mentioned above), seroprotection and seroconversion rates against the vaccine antigen were above 50 per cent in almost all cases (Figure 3). However, the use of two vaccine doses did not necessarily result in high seroconversion and seroprotection rates against heterologous antigens. Higher immunogenicity upon the administration of two vaccine doses is likely due to using two doses separated in time rather than the administration of an increased total amount of antigen. In one study, two doses of 3.75 μg of HA administrated 14 or 21 days apart resulted in the doubling of seroprotection rates (measured by HI assay) against the vaccine antigen and two heterologous antigens, as compared to one single dose of 7.5 μg of HA [21]. The present dataset also included several studies in which three or four vaccine doses were used, entailing a booster dose administered several months after the initial vaccination study. A third booster dose with an AS03 adjuvanted split inactivated vaccine resulted in an increased GMT (HI) against the vaccine antigen (179.1 to 441.0) and against a heterologous antigen (18.1 to 191.9) [44]. A significant increase in SP (HI) [28] or SC (HI) [27] was also observed against heterologous antigens tested upon administration of a third booster dose of subunit vaccines 6 or 16 months after the second dose. An increased number of doses also improved the immunogenicity and cross-reactivity of vector-based vaccines. In one study, a subunit booster dose was required after three doses of adenovirus subtype 4 vaccine to lead to detectable seroconversion [40]. In another study, a third booster dose of MVA vaccine resulted in a significant increase in GMT (HI) against the homologous and a heterologous antigen (not depicted in Figure 3 since the data could not be extracted from the article) [39]. Taken together, a third or fourth booster dose several months after an initial vaccination scheme is likely to be beneficial for the height and breadth of the immune response, but the benefit of such a vaccination strategy in the context of a pandemic scenario, when time is of the essence, might be limited. 

The majority of the high immunological endpoint values against both the vaccine and heterologous antigens were generated using split inactivated vaccines in combination with AS03 adjuvants. In several studies, the effect of split inactivated vaccines with or without AS03 was directly compared and improved immunological endpoints against both vaccine and heterologous antigens in all parameters tested were observed using the AS03 adjuvant [17,18,20,22,25]. In one study, a split inactivated vaccine was also combined with the AF03 adjuvant [16]. Interestingly, the SP (HI) and GMT (HI, VN) against both the vaccine and heterologous antigen were lower as compared to those in studies in which the AS03 adjuvant was used but with otherwise similar variables [17,18,19,20,21,22,23,24,25]. In three studies, whole inactivated vaccines were used in combination with an aluminum salt adjuvant [32,33,35], and the use of aluminum salt adjuvant was directly compared with that of no adjuvant in one study [32]. Here, the addition of aluminum salt adjuvant resulted in significantly lower SP and SC rates against the homologous and heterologous antigens as measured by MN. This observation is in line with a previously performed meta-analysis on the use of aluminum salt adjuvants in different A/H5 vaccine types, in which immunological endpoints against the vaccine antigen obtained using aluminum salt adjuvanted vaccines were inferior to those obtained using their unadjuvanted counterparts [57]. In the present dataset, subunit vaccines have been combined with either MF59 [27,28,29,30,31] or an aluminum salt adjuvant [38]. The use of MF59 adjuvant in subunit vaccines was directly compared to that of no adjuvant in one study [27], in which it resulted in at least a two-fold increase in GMT (MN) against homologous and heterologous antigens. All in all, limited combinations of vaccine types and adjuvants were assessed in the included studies. Most importantly, the AS03 adjuvant, which when combined with split inactivated vaccines resulted in high immunological endpoints after two doses, was not assessed in combination with whole inactivated, subunit or VLP vaccines. As such, it remains difficult to assess the best combination of vaccine type and adjuvant to reach high immunological endpoints against both homologous and heterologous antigens. 

It is important to take into consideration the amount of HA protein per vaccine dose necessary to generate adequate immune responses for (pre-)pandemic vaccine development, since an antigen sparing strategy can allow the generation of more vaccine doses with the existing influenza vaccine production capacity. The effect of the amount of HA protein per dose on immunological endpoints was directly assessed in six studies in which inactivated vaccines with fixed amounts of adjuvant were used [16,25,28,32,33,35]. Increasing the HA protein amount per dose resulted in significantly higher immunological endpoints against both vaccine and heterologous antigens only in two studies [16,33]. Those were performed with AF03 adjuvanted split inactivated vaccines and with aluminum salt adjuvanted whole-inactivated vaccines, which resulted in lower immunological endpoints when compared to AS03 adjuvanted split inactivated vaccines. In the other four of these studies, no significant dose-dependent effect of the amount of HA protein was observed, in dose ranges from 1.9 to 30 μg HA [25,28,32,35]. Taken together, these data suggest that in case of a A/H5 pandemic, an antigen sparing strategy can be applied if an effective vaccine type and adjuvant combination, such as AS03-split inactivated vaccines [25], is used. Additionally, for the MVA vectored vaccine [39] and the VLP vaccine [36], a dose-dependent effect was observed, in contrast to the live attenuated vaccine, where no dose-dependent effect was observed [37]. 

The timing between prime and boost vaccinations is a balancing act in a pandemic situation. Shorter intervals are favored to reach vaccination coverage faster. On the other hand, a longer time period between prime and boost vaccinations could result in an improved breadth and height of the immune response due to an increased timeframe available for B cell affinity maturation and memory B-cell production. The timing between two doses was directly compared in three studies, in which longer intervals between the doses resulted in significantly improved immune responses [21,29,33]. A 28-day interval between two doses of an adjuvanted whole inactivated vaccine resulted in a 50% increase in GMT (MN) as compared to a 14-day interval [33]. Increasing a 7-day interval to 14- or 21-day interval between two doses of an AS03 adjuvanted split inactivated vaccine resulted in an increase in SP (HI) from 74.3% to >92% against the vaccine antigen and 35.1% to >59% and 58.1% to >82% against two heterologous antigens in serum samples obtained 14 days after the last vaccination [21]. The differences were less pronounced in serum samples obtained 21 days after the vaccination. In another study, intervals of 7, 14, 21 or 42 days between two doses of MF59 adjuvanted subunit vaccines were assessed, and a positive trend was observed between the GMTs (HI) against the vaccine antigen and the time interval; however, statistical tests were not described [29]. No differences in the immunological endpoints were observed against the heterologous antigen, for which only the 14- and 21-day intervals were compared. 

### 3.2. Heterologous Vaccination

#### 3.2.1. Description of the Dataset

The vast majority of the studies on heterologous vaccination were follow-up studies of homologous prime-boost vaccination regimens and therefore entailed the administration of a total of three or four vaccine doses. Most frequently, two vaccine doses with a primary antigen (referred here as to primary vaccination(s)) were given followed by one vaccine dose with an antigenically different secondary antigen (referred here as to secondary vaccination(s)). The period between the primary vaccination(s) and secondary vaccination(s) ranged from six months to seven years. Data on immunological endpoints against only the two vaccine antigens were reported in nine articles, whereas immune responses against additional antigens besides the two vaccine strains were described in four articles.

Several different vaccine types and adjuvants were represented in the included studies on heterologous vaccination strategies (Table 2, Figure 2). In 12 studies, the vaccine types used in the primary and secondary vaccination(s) were the same. In these aforementioned studies, data entries, in which split inactivated vaccines were assessed, were the most numerous (*n* = 3), representing about half of the whole dataset, followed by those from studies on subunit (*n* = 17) and whole inactivated (*n* = 12) vaccines. Additionally, two different vaccine types were used for the primary and secondary vaccination(s) in two studies [53,54]. These vaccine types were used without adjuvant or with one of the adjuvants previously discussed. The amount of HA protein per vaccine dose varied from 3.75 μg to 90 μg, with 3.75 μg being the most frequently used. Eight different combinations of primary and secondary vaccine antigens were used (Figure 2). 

#### 3.2.2. Discussion of the Data

The heterologous prime-boost data were visualized by plotting the immunological endpoint values obtained per entry for the primary vaccine antigen (x axis) against that obtained for the secondary vaccine antigen (y axis) (Figure 4). Interactive visualizations of these data are shown in Appendix A, in which information about the vaccine type, adjuvant, dose of HA, the timing between the vaccination and the blood draw, and the antigen used in the vaccine are displayed. Data from articles in which immunological endpoints were reported against additional strains than the primary and secondary vaccine antigen are plotted in Figure 5. 

There was overall a good correspondence between the immunological endpoints against the primary and secondary vaccine antigen (Figure 4). Similar conclusions on the effect of vaccine type, adjuvant use and number of doses on the immunological endpoints against homologous and heterologous antigens can be reached with the heterologous and the homologous vaccination datasets. The highest immunological endpoints were obtained using a total of three to four vaccine doses and AS03 adjuvanted split inactivated vaccines. In three studies, the addition of AS03 adjuvant to the primary vaccination(s) resulted in improved immunological endpoints against both the primary and secondary vaccine antigens after an adjuvanted secondary vaccination [42,43,45]. In one of these studies, two unadjuvanted primary vaccinations followed by two adjuvanted secondary vaccinations resulted in lower immunological endpoints (HI) as compared to subjects which received either adjuvanted primary vaccinations or no primary vaccinations [45]. As such, the authors suggest unadjuvanted primary vaccination(s) could inhibit the subsequent response to the secondary vaccine antigen. In one study, higher immunological endpoints against the homologous antigens (HI, MN) and a heterologous antigen (MN) were obtained by adding AS03 to the secondary vaccine rather than to the primary vaccine. Of note, statistical tests were not described [47]. 

The interval between the primary and secondary vaccinations varied largely between studies. In the majority of the studies, an interval of over half a year was used, thereby mimicking a scenario in which a part of the population would be vaccinated with a pre-pandemic vaccine before the start of the pandemic and subsequently receive a secondary vaccine matched to the pandemic strain. In one study, the GMT (MN) against both homologous antigens after a single secondary vaccination performed 6 years after the primary vaccination [51] were in a similar range as those described in other studies using shorter intervals and the same vaccine type and adjuvant, suggesting that the immune memory persists for at least six years after initial vaccination(s). In another study, intervals of 6, 12 or 36 months between primary and secondary vaccinations were compared with split inactivated, AS03A adjuvanted vaccines, resulting in a total of three vaccinations per subject [45]. The GMT (HI, MN) obtained 21 days after secondary vaccination against both homologous antigens varied between groups, but inconsistently to the timing between the primary and secondary vaccinations, suggesting that similar immunological endpoints can be reached with 6- up to 36-month intervals. Shorter intervals between primary and secondary vaccinations, ranging from two to four weeks as classically performed in homologous vaccination regimens, should be addressed in future studies to consider a scenario where two different vaccine strains are available simultaneously. 

The effect of the amount of HA protein in the primary whole inactivated adjuvanted (aluminum salt) or unadjuvanted vaccine was assessed in one study [52]. The immune responses against the primary vaccine antigen after a heterologous boost increased significantly when the primary vaccine contained 30 μg HA and was adjuvanted compared to 3.75, 7.5, or 15 μg HA adjuvanted or 7.5 or 15 μg HA unadjuvanted. Interestingly, this effect was not observed in the immune responses against the secondary vaccine antigen or the heterologous antigens assessed. Moreover, this finding was in contrast to the observations after the primary vaccinations, where the 7.5 or 15 μg HA unadjuvanted vaccines resulted in the highest immunological endpoints [32]. The amount of HA in the secondary vaccine varied in one study [46]. The GMT (HI) was over fourfold higher with a secondary vaccination of 90 μg HA as compared to 15 μg HA, using an unadjuvanted split-inactivated vaccine. The observed improvement was observed against both the primary and secondary vaccine strains. 

In six studies, the cross-reactivity of the immune response against strains antigenically different from the primary and secondary vaccination strains was evaluated (Figure 5) [43,47,50,51,52,53]. In general, the higher the GMT titers were against the primary and secondary antigens, the higher they also were against heterologous antigens. Detectable immunological endpoints were observed for all experimental groups against all heterologous strains assessed. In addition, GMT (HI, MN) titers were above 1:40, the threshold used in most studies to define seroprotection, against almost all heterologous antigens (except that depicted in Figure 5E). There were no statistically significant differences between the height of the GMT against the homologous and heterologous antigens.

Due to the small size of the heterologous vaccination dataset and the heterogeneity in terms of the vaccine type, adjuvant, and timing between primary and secondary vaccination in the homologous and heterologous datasets, it was not possible to directly compare the two datasets and identify if the breadth of the antibody response was improved upon heterologous prime-boost vaccination. However, two individual articles in the present selection included a direct comparison between a homologous and heterologous vaccination strategy, both using AS03 adjuvanted split inactivated vaccines. In the first study, a heterologous boost with an A/Indonesia/05/2005 (clade 2.1) vaccine after one or two doses of a A/Vietnam/1194/2004 (clade 1) vaccine increased GMT (HI) by at least two-fold against A/Indonesia/05/2005 without reducing titers against A/Vietnam/1194/2004, as compared to a homologous boost [41]. Unfortunately, immune responses against other antigens than those used in the primary and secondary vaccination were not assessed. In the second study, a two-dose homologous vaccination with A/turkey/Turkey/1/2005 (clade 2.2) was compared to a vaccination regimen consisting of one dose of A/Indonesia/05/2005 vaccine followed by one dose of A/turkey/Turkey/1/2005 vaccine [47]. In the homologous vaccination regimen, the second vaccination was given 12 months after the first dose, whereas in the heterologous vaccination, the second vaccination was given 18 months after the first dose. In this study, the immunological endpoints were determined using the two vaccine strains, as well as six additional heterologous strains. Significant increases in GMT (HI, MN) were observed only for the A/Indonesia/05/2005 secondary vaccine antigen and related A/Indonesia/NIHRD-12379/2012 (clade 2.1.3.2) heterologous antigen, but not for the other heterologous antigens assessed for which only a positive trend was observed.

## 4. Discussion

Influenza viruses of the A/H5 GsGd lineage pose a serious threat to human health and a challenge for pandemic preparedness and vaccine design due to their ongoing evolution resulting in antigenic diversity. The present article reviewed the literature on the immunogenicity and cross-reactivity of humoral immune responses induced by pre-pandemic A/H5 vaccines upon homologous and heterologous prime-boost vaccination strategies in humans. An analysis of the available data highlighted several aspects leading to increased immunogenicity and/or cross-reactivity. In general, immunological endpoint values against the strain(s) homologous to the vaccine(s) correlated with those obtained against heterologous strains. Overall, the highest immunological endpoints observed after two vaccine doses were obtained with AS03 adjuvanted split inactivated vaccines. A similar conclusion was reached by Zheng et al. [58] upon their analysis of clinical data on the immunogenicity of H7N9 influenza vaccines. 

Our analysis was limited by several aspects of the available data from the included studies. Most importantly, the variation in design between the different studies did not allow for direct comparisons and to draw strong conclusions on the effect of single parameters, such as the effect of vaccine type or adjuvant separately. In addition, in the vast majority of the studies, the numbers of heterologous antigens tested was too limited to assess the breadth of immune responses. In a follow-up study of Kreijtz et al. [39], de Vries et al. addressed this issue by using a panel of 18 different A/H5 viruses, including viruses from contemporary circulating antigenically distinct clades, to assess (non)neutralizing induced upon vaccination with an MVA vaccine [59]. Upon a booster immunization a year after the initial one or two doses, cross-reactive antibodies that recognized the HA head of antigenically distinct A/H5 viruses were detected by protein array assay. Neutralizing antibodies against recent clade 2.2.1.2, 2.3.2.1 and 2.3.4.4. were also detected, generally leading to seroprotection, as determined by GMT (MN) higher than 40. Titers obtained using HI assay were generally lower. Another approach to assess the breadth of immune responses upon heterologous vaccination was taken in the study by Wang et al., where the IgG concentrations induced were assessed against a panel of 21 A/H5 viruses from different genetic (sub)clades, including from the ones currently circulating [60]. A modification of the antibody landscape method [61] was used for visualization and analysis of the breadth of the immune response by taking into consideration the relative antigenic differences between the A/H5 antigens assessed. The IgG responses obtained were highest against the virus used in the primary vaccination, followed by antigenically similar viruses. 

Another limitation of the clinical studies included in this analysis is that the majority of the vaccine and heterologous antigens were based on relatively old A/H5 strains isolated before 2008. Since then, viruses belonging to novel A/H5 genetic subclades have emerged with distinct antigenic properties from the vaccine strains used to date in clinical studies [7]. It therefore remains unknown if the vaccination regimens presented in this analysis would induce humoral immune responses to protect against currently circulating A/H5 GsGd strains. To assess the breadth of the immune response elicited upon vaccination against more recent strains, human sera obtained from clinical studies should be evaluated against a selected panel of viruses representing the current antigenic diversity of A/H5 GsGd viruses. In addition, the development of A/H5 vaccines based on more recent A/H5 viruses is warranted, including viruses with NA subtypes beyond N1 (N2, N6, N8) given that these viruses are now dominating.

Data obtained beyond 100 days after the last vaccine dose administration were limited and very variable. Therefore, immune response longevity was not addressed in our primary analysis. However, information on the persistence of the immune responses upon vaccination is essential to assess (i) the possibility to prime (part of) the population before the onset of a pandemic (pre-pandemic vaccination) and (ii) the need for boosting during the pandemic, as a part of the population might be exposed to the virus within months after the onset of the pandemic. To this end, the data on longevity of the immune response beyond 100 days after administration available from the current datasets were visualized in Appendix A. In general, SC and SP values were maintained after 112 to 183 days upon homologous prime-boost vaccination but had decreased after about a year [26]. This decrease was less pronounced for the data obtained by MN as compared to HI, which might partially be due to the use of lower thresholds in MN [26]. Interestingly, trends were similar whether immunological endpoints against the vaccine antigen or heterologous antigens were considered. Gillard et al. [45] performed an extensive analysis of the longevity of the immune response upon heterologous prime-boost vaccination strategy. Immunological endpoints obtained at 6, 12, 18, 24, 30, 36, 42, and 48 months after the primary vaccination were reported. There was a gradual reduction in the GMT (HI and MN) and SP (HI) observed over time, which was comparable against the primary and secondary strain with all intervals of the secondary vaccination. The SP (HI) was reduced from >90% to >40% comparing 21 days to 3 years after secondary vaccination (data not shown in Appendix A). The SP (MN) did not decrease over time, probably due to a lower threshold used to define SP (>1:28 rather than >1:40), as both GMT assessed by HI and MN decreased similarly. However, data from Gillard et al. suggest long-lived immune responses can be induced by A/H5 vaccines. It is unclear whether the longevity of the response observed in this study was due to the heterologous vaccination strategy, since the direct comparison with a homologous vaccination strategy was not performed.

The current correlates of protection used in clinical studies on A/H5 viruses are based on data accumulated over decades on seasonal influenza viruses, and it remains unclear how well these can be extrapolated to A/H5 or other zoonotic influenza viruses. Correlates of protection against seasonal influenza viruses are determined in the context of significant pre-existing immunity in the human population, while such pre-existing immunity against A/H5 or other avian influenza viruses is low or inexistent. Given that assessing correlates of protection in a pre-pandemic context is impossible, animal studies can aid to assess the protective effect of A/H5 vaccination strategies and relate immunological parameters to protection from disease. The results of preclinical studies on A/H5 vaccines in ferrets are generally in line with data from clinical trials. In the ferret model, the presence of an adjuvant and the administration of two or more vaccine doses resulted in improved immunological endpoints and improved survival rates and clinical outcomes upon challenge with A/H5 viruses [62,63]. Moreover, vaccination with an A/H5 antigen heterologous to the challenge virus generally resulted in an improved survival rate and clinical outcomes in ferrets as compared to unvaccinated animals [64,65,66,67], although to a lesser degree than in ferrets vaccinated with a homologous antigen [64,65]. Interestingly, reduced disease severity was also observed upon homologous or heterologous vaccination resulting in HI titers below 40 or even undetectable [65,68,69]. Thus, pre-clinical studies highlight the need to develop robust and more sensitive assays to detect neutralizing antibodies, but also non-neutralizing and cellular responses to determine correlates of protection beyond an HI titer higher than 40. Animal models to study vaccine efficacy should be further developed and standardized to facilitate the extrapolation of pre-clinical to clinical data. One critical point is the lack of tools for immunological studies in species other than mice. As the ferret is regarded to be the animal model of choice to study influenza viruses, there is an urgent need to develop novel tools to study the ferret immune responses.

The present analysis included studies with several vaccine types and adjuvants, but only few combinations of those were addressed. For example, the AS03 adjuvant resulted in promising immunological endpoints in combination with split inactivated vaccines but was not tested in combination with whole inactivated vaccines, although there is an indication that unadjuvanted whole inactivated vaccines might be more immunogenic than split inactivated vaccines in an unprimed population [70]. In two studies, which were not included in our dataset, the combination of either split inactivated [71] or whole inactivated vaccines [72] with an aluminum phosphate adjuvant was assessed. The SP (HI) values obtained against the vaccine antigen were 30% after one dose and 60% after two doses in the former study and 80% after one dose in the latter study. In another study, which was not included in our dataset, the addition of MF59 adjuvant to a subunit vaccine resulted in a significant increase in GMT (HI, MN) against the homologous antigen but in the breadth of the response as determined by SP (HI) [73]. Taken together, these examples highlight the need to systematically assess different combinations of vaccine type and adjuvant. Moreover, approaches using inactivated vaccines have focused on only eliciting humoral responses, and it has been established that inactivated vaccines inefficiently elicit cellular immune responses, especially CD8+ T-cell responses, as assessed with seasonal unadjuvanted trivalent split-inactivated vaccines [74]. Attention should be focused on other vaccine platforms that allow de novo synthesis of viral proteins necessary to activate the cellular arm of the immune system, such as vector-based vaccines and mRNA vaccines, which have gained particular interest due to the promising results obtained with SARS-CoV-2 [75]. MRNA vaccines might also offer an advantage in a pandemic situation compared to virus-based vaccines since they do not rely on the generation of vaccine virus seed stocks. Broad B-cell and T-cell responses could also be achieved by combining vaccine platforms [76]. Intranasally administered (inactivated) vaccines could offer the advantage of inducing neutralizing nasal IgA antibodies besides serum IgG antibodies [77]. Standardization of non-HI assays, such as determining levels of antibodies that mediate antibody-dependent cellular cytotoxicity and antibody cellular phagocytosis, direct virus neutralization assays, and T-cell assays would allow more direct comparisons between different clinical studies and help to develop and license novel vaccine platforms [78].

Research on mRNA vaccines against influenza viruses remains limited. Results from a clinical study in which mRNA vaccines against avian influenza H7N9 and H10N8 viruses were used were promising. The most optimal doses administrated in two-dose vaccination series 3 weeks apart resulted in seroprotection rates of 96.3 % and 96.6% against H7N9 and 100% and 87% against H10N8, as measured by HI assay (titers 1 >= 40) or MN (titers >= 20) assay, respectively [79]. To our knowledge, no clinical data on A/H5 mRNA vaccines have been reported to date. However, A/H5 mRNA vaccines resulted in the protection of mice from a lethal infection with the virus strain homologous to the vaccine [80]. Taken together, these results provide a promising foundation for clinical studies on mRNA vaccines for A/H5 viruses and other avian influenza subtypes. 

Heterologous prime-boost vaccination strategies have been put forward in recent years as a way to (i) elicit immune responses against two antigenically different antigens (primary and secondary antigens), (ii) boost immune responses against common epitopes that might be shared with other antigenically distinct strains, and (iii) decrease “antigen trapping” compared to homologous prime-boost vaccination, during which the binding of matched antibodies elicited upon the prime might decrease the antigen load available in the boost. All these aspects would potentially contribute to an increased breadth of immune responses. The data obtained in two clinical studies [41,47] in which a direct comparison was made between homologous and heterologous vaccination strategies suggest broader immune responses with the latter approach. However, underlying immune mechanisms of potential increased breadth upon heterologous prime-boost vaccination remain to be elucidated. To gain mechanistic insights, immune responses should also be analyzed at the monoclonal level. Further studies are warranted to also address aspects such as the optimal timing between the prime and the boost, antigenic relatedness between the two vaccine antigens and the order of the prime and the boost. The order of the prime and boost should be carefully investigated in the context of immunological imprinting or original antigenic sin, according to which overall immune responses would be shaped by those against the prime vaccination [81]. A way to avoid the detrimental effects of immunological imprinting might be to administer the two heterologous vaccine antigens in a bivalent formulation yet compromising the potential of boosting immune responses against shared epitopes. The optimal range of antigenic difference between the two strains in a heterologous prime-boost vaccination to result in optimal immune responses needs to be determined. Should the antigenic difference between the two vaccine strains be too small, there would be no added value of heterologous over homologous prime-boost. On the other hand, should the antigenic difference between the vaccine strains be too large, the lack of shared immunodominant epitopes might lead to boosting of immune responses against more conserved yet less immunogenic viral epitopes. All in all, an improved understanding of the concept of heterologous prime-boost vaccinations would allow to fully assess its potential suitability for use in a (pre-)pandemic context. 

Although important aspects remain to be studied in more detail, clinical studies on A/H5 vaccines have provided important pieces of information for A/H5 GsGd pandemic preparedness. 

## Figures and Tables

**Figure 1 vaccines-09-01465-f001:**
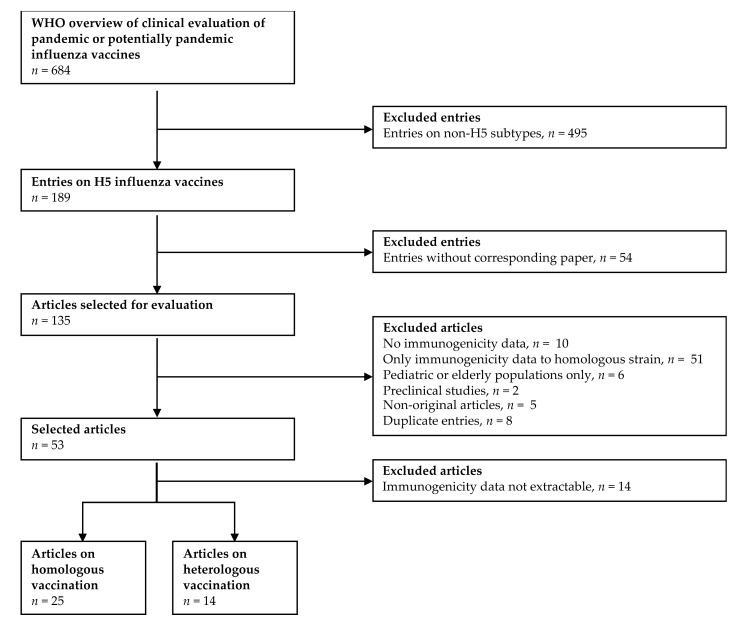
Flowchart summarizing the data selection in present analysis. The WHO overview of clinical evaluation of pandemic or potentially pandemic influenza vaccines [15] was used as a starting point for data selection.

**Figure 2 vaccines-09-01465-f002:**
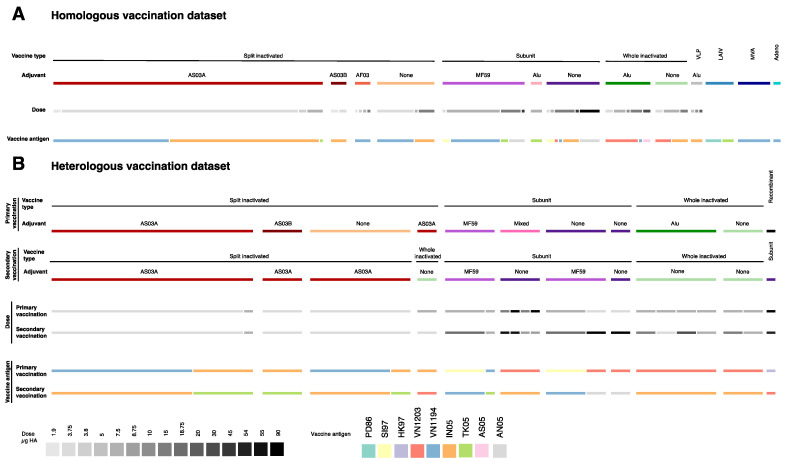
Overview of present dataset. (**A**) Homologous vaccination dataset; (**B**) heterologous vaccination dataset. For each, the width of the bars represents the number of entries in each dataset with the reported characteristics (vaccine type, adjuvant use, amount of HA or vaccine strain). The homologous dataset contained 162 entries and the heterologous dataset contained 60 entries. The ‘mixed’ adjuvant refers to a pooled experimental group where subjects received either aluminum salt adjuvanted, MF59 adjuvanted or unadjuvanted vaccines. Abbreviations: HA: hemagglutinin; PD86: A/Duck/Potsdam/1402-6⁄1986; SI97: A/duck/Singapore/1997; HK97: A/Hong Kong/156/1997; VN1203: A/Vietnam/1203/2004; VN1194: A/Vietnam/1194/2004; IN05: A/Indonesia/05/2005; TK05: A/turkey/Turkey/1/2005; AS05: A/Chicken/Astana/6/2005; AN05: A/Anhui/1/2005.

**Figure 3 vaccines-09-01465-f003:**
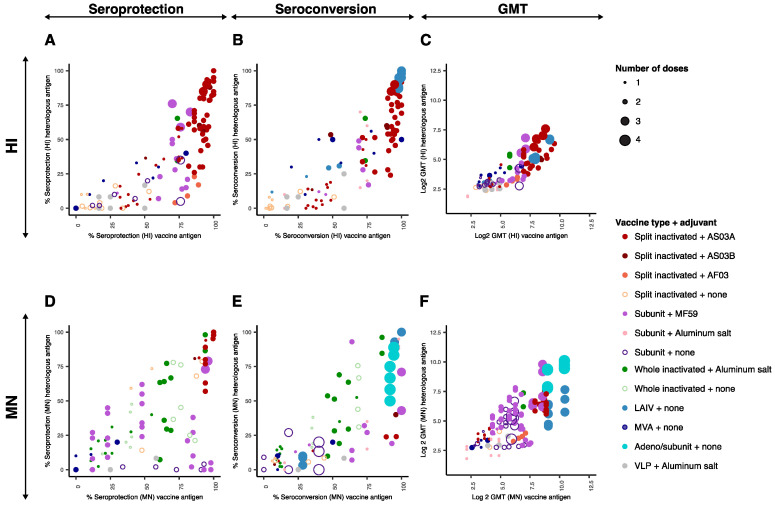
Immunological endpoints upon homologous vaccination regimens. Immunological endpoint values obtained against the homologous antigen (x-axis) are plotted against those obtained against heterologous antigens (y-axis). For clarity, the confidence intervals were not displayed. The color and fill of the individual points indicate the vaccine type and adjuvant combination according to the legend. The size of the circles indicates the number of vaccine doses (one to four) used before the serum sample was obtained according to the legend. (**A**) Percentage of seroprotected subjects measured by HI assay; (**B**) percentage of seroconverted subjects measured by HI assay; (**C**) Log2 of the geometric mean HI titer; (**D**) percentage of seroprotected subjects measured by MN assay; (**E**) percentage of seroconverted subjects measured by MN assay, (**F**) Log2 geometric of the mean MN titer. Abbreviations: HI: hemagglutination inhibition; MN: microneutralization; GMT: geometric mean titer; LAIV: live attenuated influenza vaccine; MVA: modified vaccinia virus Ankara; VLP: virus-like particle.

**Figure 4 vaccines-09-01465-f004:**
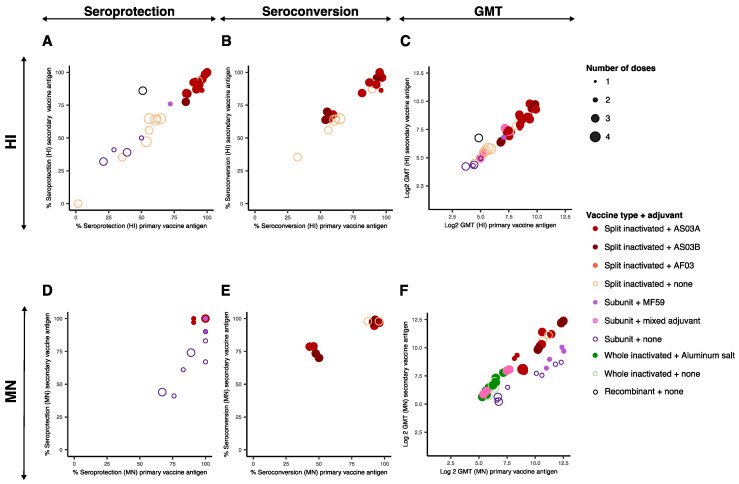
Immunological endpoints of heterologous vaccination strategies. Immunological endpoint values obtained against the antigen homologous to the primary vaccine strain (x-axis) are plotted against those obtained against the antigen homologous to the secondary vaccine strain (y-axis). Since the vast majority of experimental groups received heterologous vaccination(s) with the same vaccine type as the primary vaccination, the color and fill of the individual points indicate the vaccine type and adjuvant combination used in the primary vaccination according to the legend. The size of the circle indicates the number of vaccine doses (one to four) used before the serum sample was obtained as indicated on the right side of the figure. Data on persistence of the immune response was excluded. (**A**) Percentage seroprotected subjects measured by HI assay; (**B**) percentage of seroconverted subjects measured by HI assay; (**C**) Log2 of the geometric mean HI titer; (**D**) percentage of seroprotected subjects measured by MN assay; (**E**) percentage of seroconverted subjects measured by MN assay; (**F**) Log2 of the geometric mean MN titer. Abbreviations: HI: hemagglutination inhibition; MN: microneutralization; GMT: geometric mean titer.

**Figure 5 vaccines-09-01465-f005:**
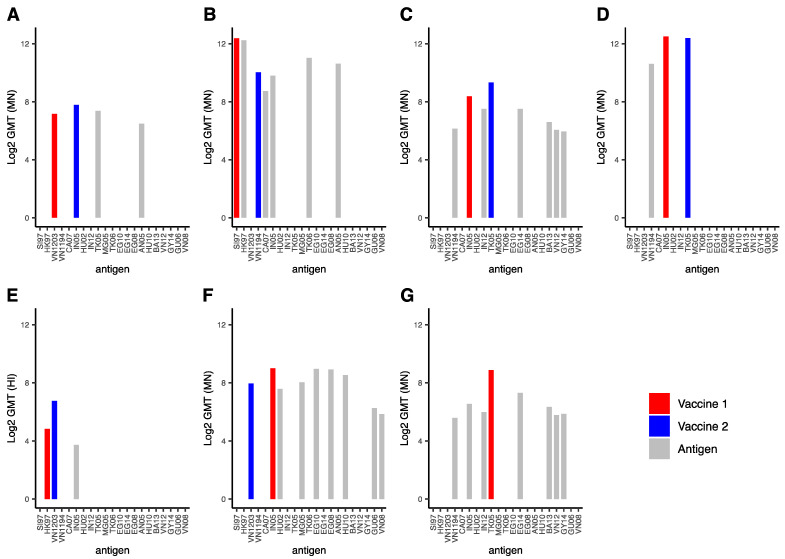
Geometric mean titers obtained against various antigens after homologous or heterologous vaccination regimens. The data in each panel were derived from one of the six articles in which a heterologous vaccination regime was assessed and immunological endpoints against one or multiple antigens heterologous to the vaccine antigens were reported. For each study, the experimental group and time points yielding the highest geometric mean titers are displayed. In case immunological endpoints were not assessed against a particular antigen, no bar is depicted. Bars indicating titers against the primary vaccine, secondary vaccine and heterologous strains are red, blue and grey respectively. (**A**) data obtained from Erlich et al., 2009 [48]; (**B**) data obtained from Khurana et al., 2014 [47]; (**C**,**G**) data obtained from Levine et al., 2019 [44]; (**D**) data obtained from Risi et al., 2011 [40]; (**E**) data obtained from Goji et al., 2008 [49]; (**F**) data obtained from Haveri et al., 2021 [54]. In panel (**G**), the homologous vaccination regime counterpart of panel (**C**) is displayed for direct comparison. Abbreviations: GMT: geometric mean titer, MN: microneutralization, SI97: A/duck/Singapore/1997 (clade C), HK97: A/Hong Kong/156/1997 (clade 0), VN1203: A/Vietnam/1203/2004 (clade 1), VN1194: A/Vietnam/1194/2004 (clade 1), CA07: A/Cambodia/R0405050/2007 (clade 1), IN05: A/Indonesia/05/2005 (clade 2.1), HU02: A/duck/Hunan/795/2002 (clade 2.1.1), IN12: A/Indonesia/NIHRD-12379/2012 (clade 2.1.3.2), TK05: A/turkey/Turkey/1/2005 (clade 2.2), MG05: A/whooper-swan/Mongolia/244/2005 (clade 2.2), TK06: A/Turkey/15/2006 (clade 2.2), EG10: A/Egypt/N03072/2010 (clade 2.2.1), EG14: A/Egypt/N04915/2014 (clade 2.2.1), EG08: A/Egypt/3300-Namru3/2008 (clade 2.2.1.1), AN05: A/Anhui/1/2005 (clade 2.3.4), HU10: A/Hubei/1/2010 (2.3.2.1a), BA13: A/duck/Bangladesh/19097/2013 (clade 2.3.2.1a), VN12: A/duck/Vietnam/NCVD-1584/2012 (clade 2.3.2.1c), GY14: A/gyrfalcon/Washington/410886/2014 (clade 2.3.4.4), GU06: A/goose/Guiyang/337/2006 (clade 4), VN08: A/chicken/Vietnam/NCVD-016/2008 (clade 7.1).

**Table 1 vaccines-09-01465-t001:** Overview of the articles selected for the homologous prime-boost vaccination analysis.

Study	Vaccine Type	Adjuvant	Vaccine Strain (Clade)	Dose (μg HA) ^1^	Number of Doses	Interval between Doses (Days) ^1^	Heterologous Strains (Clade) Used for Immunological Endpoint(s)
Levie et al., 2008 [16]	Inactivated split virus	AF03	VN1194 (1)	1.9, 3.75, 7.5, 15	2	21	IN05 (2.1)
Chu et al., 2009 [17]	Inactivated split virus	AS03, none	VN1194 (1)	3.75	2	21	IN05 (2.1)
Langley et al., 2010 [18]	Inactivated split virus	AS03A, AS03B, none	IN05 (2.1)	3.75	2	21	VN1203 (1)
Nagai et al., 2010 [19]	Inactivated split virus	AS03	IN05 (2.1)	3.75	2	21	VN1203 (1), TK05 (2.2)
Chu et al., 2011 [20]	Inactivated split virus	AS03, none	VN1194 (1)	3.75	2	21	IN05 (2.1)
Lasko et al., 2011 [21]	Inactivated split virus	AS03	IN05 (2.1)	3.75, 7.5	2	7, 14 or 28	VN1203 (1), TK05 (2.2)
Thongcharoen et al., 2011 [22]	Inactivated split virus	AS03	VN1194 (1)	3.75	2	21	IN05 (2.1)
Yang et al., 2012 [23]	Inactivated split virus	AS03	IN05 (2.1)	3.75	2	21	VN1203 (1)
Izurieta et al., 2015 [24]	Inactivated split virus	AS03	IN05 (2.1)	3.75	2	21	VN1203 (1)
Naruse et al., 2015 [25]	Inactivated split virus	AS03	IN05 (2.1)	1.9, 3.75, 7.5	2	21	VN1194 (1), AN05 (2.3.4), QI05 (2.2)
Schuind et al., 2015 [26]	Inactivated split virus	AS03A, AS03B, none	IN05 (2.1)	1.9, 3.75, 15	2	21	VN1203 (1)
Stephenson et al., 2005 [27]	Inactivated subunit	MF59, none	SI97 (C) ^2^	7.5, 15, 30 ^3^	2 + 1	21, + 16 months	HK97 (0)
Banzhof et al., 2009 [28]	Inactivated subunit	MF59	VN1194 (1)	7.5, 15	2 + 1	21, + 6 months	TK05 (2.2)
Beran et al., 2010 [29]	Inactivated subunit	MF59	VN1194 (1)	7.5	2	7, 14, 21 or 42 ^4^	TK05 (2.2)
Bihari et al., 2012 [30]	Inactivated subunit	MF59	TK05 (2.2)	7.5	2	21	VN1194 (1), IN05 (2.1)
Vesikari et al., 2012 [31]	Inactivated subunit	MF59	VN1194 (1)	7.5	2	21	VN1194 (1), TK05 (2.2)
Ehrlich et al., 2008 [32]	Whole inactivated	Aluminum salt, none	VN1203 (1)	3.75, 7.5, 15, 30	2	21	HK97 (0), IN05 (2.1)
Wu et al., 2009 [33]	Whole inactivated	Aluminum salt	VN1194 (1)	5, 10, 15	2	14 or 28	IN05 (2.1), TK05 (2.2), AN05 (2.3.4)
Tambyah et al., 2012 [34]	Whole inactivated	None	IN05 (2.1)	3.75, 7.5	2	21	VN1203 (1)
Sansyzbay et al., 2013 [35]	Whole inactivated	Aluminum salt	AS05 (2)	7.5, 15	1	NA	VN1203 (1)
Landry et al., 2010 [36]	VLP	Aluminium salt	IN05 (2.1)	5, 10, 20	2	21	VN1194 (1), TK05 (2.2), AN05 (2.3.4)
Rudenko et al., 2008 [37]	Live attenuated	NA	PD86 (C)	10^6.9^, 10^8.3^ TCID50	2	21	IN05 (2.1)
Pitisuttithum et al., 2017 [38]	Live attenuated + inactivated subunit	NA + Aluminum salt	TK05 (2.2)	8 logEID + 7.5 µg HA	2	21	TH04 (1), IN05 (2.1), LA07 (2.3.4)
Kreijtz et al., 2014 [39]	MVA-vectored	NA	VN1194 (1)	10^7^ PFU, 10^8^ PFU	1/2 + 1	21 + 12 months	IN05 (2.1), NL14 (2.3.4.4)
Khurana et al., 2015 [40]	Ad4-vectored vaccine + inactivated subunit	NA	VN1194 (1)	10^7^, 10^11^ particles + 90 µg HA	3 + 1	56 + 3–12 months	IN05 (2.1), TK05 (2.2), EG10 (2.2.1), AN05 (2.3.4)

^1^ Unless noted otherwise; ^2^ C: classical H5, non-GsGd lineage; ^3^ data obtained with three different HA doses were combined into one group because no effect of amount of HA on immunological endpoints was observed; ^4^ data from 7 and 42 days only reported against homologous antigen. Abbreviations: VLP: virus-like particle, MVA: modified vaccinia virus Ankara, Ad4: adenovirus subtype 4, TCID50: tissue culture infective dose 50%, EID: egg infective dose, PFU: plaque forming units, PD86: A⁄Duck⁄Potsdam⁄1402-6⁄1986, SI97: A/duck/Singapore/1997, HK97: A/Hong Kong/156/1997, VN1203: A/Vietnam/1203/2004, VN1194: A/Vietnam/1194/2004, TH04: A/Thailand/16/2004, IN05: A/Indonesia/05/2005, TK05: A/turkey/Turkey/1/2005, QI05: A/bar-headed-goose/Qinghai/1A/2005, EG10: A/Egypt/N03072/2010, AN05: A/Anhui/1/2005 (clade 2.3.4), LA07: A/Laos/Nong Khai/1/2007, NL14: A/chicken/Netherlands/EMC-3/2014.

**Table 2 vaccines-09-01465-t002:** Overview of the articles selected for the heterologous prime-boost vaccination analysis.

Study	Primary Vaccine Type	Secondary Vaccine Type	Primary Vaccine Adjuvant	Secondary Vaccine Adjuvant	Primary Vaccine Strain (Clade)	Secondary Vaccine Strain (Clade)	Number of Doses	Interval (Months) ^1^	Heterologous Strains (Clade) Used for Immunological Endpoint(s)
Schwarz et al., 2009 [41] ^2^	Inactivated split virus	Inactivated split virus	AS03A	AS03A	VN1194 (1)	IN05 (2.1)	1/2 + 1	6	
Leroux-Roels et al., 2010 [42]	Inactivated split virus	Inactivated split virus	AS03A, none	AS03A	VN1194 (1)	IN05 (2.1)	2 + 1/2	14	
Risi et al., 2011 [43]	Inactivated split virus	Inactivated split virus	AS03A, AS03B, none	AS03A, none	IN05 (2.1)	TK05 (2.2)	2 + 1	15	VN1194 (1)
Gillard et al., 2013 [44] ^2^	Inactivated split virus	Inactivated split virus	AS03A	AS03A	VN1203 (1)	IN05 (2.1)	1/2 + 1	12	
Gillard et al., 2014 [45]	Inactivated split virus	Inactivated split virus	AS03A, none	AS03A	VN1203 (1)	IN05 (2.1)	2 + 1	6, 12 or 36	
Winokur et al., 2015 [46] ^2^	Inactivated split virus	Inactivated split virus	Aluminum salt, MF59, none ^3^	None	VN1203 (1)	IN05 (2.1)	2 + 1	1.4–3.7 years	
Levine et al., 2019 [47] ^2^	Inactivated split virus	Inactivated split virus	AS03A	AS03A	IN05 (2.1)	TK05 (2.2)	1 + 1	6, 18	VN1194 (1), IN12 (2.1.3.2), EG14 (2.2.1), BA13 (2.3.2.1a), VN12 (2.3.2.1c), GY14 (2.3.4.4)
Lopez et al., 2013 [48] ^2^	Inactivated subunit	Inactivated subunit	MF59	MF59	VN1194 (1)	TK05 (2.2)	1 + 1	12	
Belshe et al., 2014 [49] ^2^	Inactivated subunit	Inactivated subunit	None	MF59, none	VN1194 (1)	AN05 (2.3.4)	1/2 + 1	24	
Galli et al., 2014 [50] ^2^	Inactivated subunit	Inactivated subunit	MF59, none	MF59	SI97 (C)	VN1194 (1)	1 + 2	6 years	HK97 (0), CA07 (1), IN05 (2.1), TK06 (2.2), AN05 (2.3.4)
Khurana et al., 2014 [51] ^2^	Inactivated subunit	Inactivated subunit	MF59, none	MF59	SI97 (C)	VN1194 (1)	1 + 2 ^4^	6 years	HK97 (0), CA07 (1), IN05 (2.1), TK06 (2.2), AN05 (2.3.4)
Ehrlich et al., 2009 [52]	Whole inactivated	Whole inactivated	Aluminum salt, none	None	VN1203 (1)	IN05 (2.1)	2 + 1	12–17	TK05 (2.2), AN05 (2.3.4)
Goji et al., 2008 [53] ^2^	Recombinant protein	Inactivated split virus	None	None	HK97 (0)	VN1203 (1)	2 + 1	7 years	IN05 (2.1)
Haveri et al., 2021 [54]	Inactivated split virus	Whole inactivated	AS03A	None	IN05 (2.1)	VN1194 (1)	2 + 2	24	

^1^ Interval between primary and secondary vaccinations in months (unless noted otherwise); ^2^ these articles also contained data on homologous vaccinations which were included in the homologous vaccination dataset; ^3^ subjects with adjuvanted and unadjuvanted primary vaccination were combined into one experimental group; ^4^ although two secondary vaccinations were performed, only the data after a single secondary vaccination was reported in the article. Abbreviations: Nb: Number, SI97: A/duck/Singapore/1997, HK97: A/Hong Kong/156/1997, VN1203: A/Vietnam/1203/2004, VN1194: A/Vietnam/1194/2004, CA07: A/Cambodia/R0405050/2007, IN05: A/Indonesia/05/2005, IN12: A/Indonesia/NIHRD-12379/2012, TK05: A/turkey/Turkey/1/2005, EG14: A/Egypt/N04915/2014, AN05: A/Anhui/1/2005, BA13: A/duck/Bangladesh/19097/2013, VN12: A/duck/Vietnam/NCVD-1584/2012, GY14: A/gyrfalcon/Washington/410886/2014.

## Data Availability

Data is contained within the article or Appendix A.

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
