# Peer review of "Cross-Reactivity Conferred by Homologous and Heterologous Prime-Boost A/H5 Influenza Vaccination Strategies in Humans: A Literature Review"

_vaccines, 2021, doi:10.3390/vaccines9121465_

Round 1

Reviewer 1 Report

In this manuscript by Kok, Fouchier, and Richard, the authors perform a literature review to understand humoral immunity against H5 viruses in humans enrolled in various clinical trials. The topic of the manuscript is important and the authors have performed a thorough literature review. Moreover, the paper is well written and easy to follow. The authors adequately describe their methods and provide two very nice supplemental interactive figures to tease apart the different parameters assessed. Overall, I only suggest a few more analyses and discussion points to improve the overall cohesiveness and thoroughness of the manuscript.

Major Comments

  1. Line 63-65 – the authors imply that seasonal influenza vaccines are the gold standard for immunogenicity against influenza viruses, which is far from accurate. It is important to note that the H5 are less immunogenic, but that the seasonal vaccines have limited immunogenicity is warranted.
  2. The authors excluded analysis of time points past 100 days, which is only slightly over 3 months post vaccination. Currently, most people receive a seasonal influenza vaccine about 3 months before the actual influenza season. Similarly, in the event of an H5 pandemic, a naïve population are likely to be exposed over the next coming year(s), not within the first several months after vaccination. Therefore, looking at antibody responses at these later time points (when available) are important for understanding the persistence of neutralizing antibodies induced by these vaccine formulations.
  3. The authors mentioned they excluded studies in pediatric and elderly populations, but what were the general demographic makeup of the studies, including age, sex, and country of origin of the study?

Minor Comments

  1. In the abstract at line 19-20, the authors mention the several aspects. It would be good to list these here and maintain the caveats that direct comparisons are limited.
  2. Line 232 – please state it as aluminum salt adjuvant.
  3. Can the authors say anything about the relative roles of MF59 versus AS03 adjuvants? They are both squalene-based adjuvants and comparison would be of interest to readers.
  4. Line 260 – related to dose sparring, is the adjuvant concentration also reduced or just the HA content?
  5. The authors have done a great job describing the clades of the HA antigens. A similar discussion of the how the different vaccine formulations (inactivated, subunit, AS03, MF59, etc.) have been shown to work would be insightful. Notably, the authors are encouraged to discuss the data found with H5 relates to similar vaccine platforms against other influenza A virus subtypes.
  6. Line 521-523 – these studies should be cited here.

Author Response

We thank the reviewers for their very constructive comments. Please find below the comments of the reviewers reproduced and our answer to each of them in red font.

In this manuscript by Kok, Fouchier, and Richard, the authors perform a literature review to understand humoral immunity against H5 viruses in humans enrolled in various clinical trials. The topic of the manuscript is important and the authors have performed a thorough literature review. Moreover, the paper is well written and easy to follow. The authors adequately describe their methods and provide two very nice supplemental interactive figures to tease apart the different parameters assessed. Overall, I only suggest a few more analyses and discussion points to improve the overall cohesiveness and thoroughness of the manuscript.

Major Comments

  1. Line 63-65 – the authors imply that seasonal influenza vaccines are the gold standard for immunogenicity against influenza viruses, which is far from accurate. It is important to note that the H5 are less immunogenic, but that the seasonal vaccines have limited immunogenicity is warranted.

Thanks for this comment. We agree with the reviewer that the efficacy of seasonal influenza vaccines is far from optimal, but this is mainly the case when there is an antigenic mismatch between vaccine strains and circulating strains or when there are changes due to the propagation of the vaccine seed strain in eggs for vaccine production leading to phenotypic changes in the hemagglutinin, but not necessarily because of low immunogenicity of seasonal influenza vaccines. Human influenza viruses are in general more immunogenic than avian influenza viruses, which is reflected by the fact that one dose of unadjuvanted vaccine is sufficient to elicit seroprotection against matched human influenza viruses, whereas at least two doses of adjuvanted vaccines are necessary to achieve similar outcome against matched avian influenza viruses.

  1. The authors excluded analysis of time points past 100 days, which is only slightly over 3 months post vaccination. Currently, most people receive a seasonal influenza vaccine about 3 months before the actual influenza season. Similarly, in the event of an H5 pandemic, a naïve population are likely to be exposed over the next coming year(s), not within the first several months after vaccination. Therefore, looking at antibody responses at these later time points (when available) are important for understanding the persistence of neutralizing antibodies induced by these vaccine formulations.

We thank the reviewer for this comment. The main reason for not including data obtained after 100 days post-vaccination was because 70% of the data in our dataset was obtained up to 100 days after the last vaccine dose. Additionally, the data obtained at later points are dispersed very sparsely and therefore we decided to not compare data from time points beyond 100 days with those obtained more recently after vaccination. However, we agree with the reviewer’s point regarding the importance of looking at antibody responses at later time points. Therefore, we have now explained in more detail the decision for not including the data beyond 100 days in line 136-142 of the Methods.  Moreover, we have included the data beyond 100 days in Supplementary Figure 3 and discussed it lines 138-145 and 450-475.

  1. The authors mentioned they excluded studies in pediatric and elderly populations, but what were the general demographic makeup of the studies, including age, sex, and country of origin of the study?

Information about the demographic makeup of each study is now included in the Supplementary Tables.

Minor Comments

  1. In the abstract at line 19-20, the authors mention the several aspects. It would be good to list these here and maintain the caveats that direct comparisons are limited.

We agree with the reviewer that adding more information about which aspects were identified in the abstract would be more informative. However, due to word count limitation, we will not be able to extend it with detailed information.

  1. Line 232 – please state it as aluminum salt adjuvant.

This was changed accordingly.

  1. Can the authors say anything about the relative roles of MF59 versus AS03 adjuvants? They are both squalene-based adjuvants and comparison would be of interest to readers.

We thank the reviewer for this comment. Unfortunately, these adjuvants have not been tested side by side in any of the studies in our analysis. Due to the variation in other parameters in the studies using MF59 or AS03, it is not possible to conclude about the relative roles of the two adjuvants when comparing the data on these adjuvants. This limitation was already addressed in the discussion, lines 514-528.

  1. Line 260 – related to dose sparring, is the adjuvant concentration also reduced or just the HA content?

Only the HA content was varied in these comparisons. This has now been clarified in the text line 261.

  1. The authors have done a great job describing the clades of the HA antigens. A similar discussion of the how the different vaccine formulations (inactivated, subunit, AS03, MF59, etc.) have been shown to work would be insightful. Notably, the authors are encouraged to discuss the data found with H5 relates to similar vaccine platforms against other influenza A virus subtypes.

We thank the reviewer for this comment. The comparison to the findings regarding this for H7N9 vaccines are now briefly mentioned in line 425-427. It does not fit the scope of our review to elaborate on the findings of other subtypes.

  1. Line 521-523 – these studies should be cited here.

References to the two studies were added as suggested.

Reviewer 2 Report

I read with interest the manuscript submitted to me for revision which I consider worthy of publication if some points are clarified.

Influenza pandemics are certainly a threat to humans and the need for an alert on the leap of species is certainly a topic of discussion, as demonstrated by the SARS-CoV-2 pandemic unfortunately still underway.

I have two observations to make:

1) it would be interesting to know the number of subjects vaccinated in the various studies (to be included in Tables 1 and 2);

2) it is not clear which cut-off is used to define both seroconversion and seroprotection.

Author Response

I read with interest the manuscript submitted to me for revision which I consider worthy of publication if some points are clarified.

Influenza pandemics are certainly a threat to humans and the need for an alert on the leap of species is certainly a topic of discussion, as demonstrated by the SARS-CoV-2 pandemic unfortunately still underway.

I have two observations to make:

1) it would be interesting to know the number of subjects vaccinated in the various studies (to be included in Tables 1 and 2);

The number of subjects vaccinated in each study is now included in Supplementary tables 1 and 2.

2) it is not clear which cut-off is used to define both seroconversion and seroprotection.

The cut-offs for seroconversion and seroprotection are already described in the methods section, lines 125-136. In our opinion, these are clearly and elaborately discussed.

Reviewer 3 Report

This is an interesting approach and a very interesting analysis of responses to the influenza virus vaccines. I believe this will be very relevant to the vaccine field in general as well as those involved with design of influenza vaccine protocols.

The paper is generally well-prepared.

Line 79 – I believe it would be good to add a very brief description of how increased immunogenicity is defined.

Here, we review the literature available on the breadth of antibody responses induced by A/H5 influenza virus vaccines in humans upon homologous and heterologous prime-boost vaccinations to identify aspects correlating with increased immunogenicity and cross-reactivity.

Line 147 – check the format of the sub-headings to clarify separation.

Line 491/reference 73 – the lack of CD-8 responses may also be due to formulation and adjuvant choices as well. It would help to comment on the formulations assessed in reference 73.

Figure 2 – This is a very good representation of the formulations.

Line 514 – “strategies have”

Line 532 – “administer”

Line 540 – Suggested:

All in all, an improved understanding of the concept of heterologous prime-boost vaccinations would allow full assessment of potential suitability for use in a (pre-)pandemic context.

Figures 3 and 4 - The description and discussion of these figures could be improved and summarized more clearly. It was difficult to follow in the text. The figure legends are clear, however.

Author Response

This is an interesting approach and a very interesting analysis of responses to the influenza virus vaccines. I believe this will be very relevant to the vaccine field in general as well as those involved with design of influenza vaccine protocols.

The paper is generally well-prepared.

Line 79 – I believe it would be good to add a very brief description of how increased immunogenicity is defined.

Here, we review the literature available on the breadth of antibody responses induced by A/H5 influenza virus vaccines in humans upon homologous and heterologous prime-boost vaccinations to identify aspects correlating with increased immunogenicity and cross-reactivity.

We thank the reviewer for this comment. We actually realized upon the reviewer’s comment that using the term high immunogenicity rather than increased immunogenicity was more appropriate. A brief description of how high immunogenicity was defined was added, as suggested.

Line 147 – check the format of the sub-headings to clarify separation.

The format of the sub-headings was double-checked and adapted where necessary to be consistent with the journal’s template.

Line 491/reference 73 – the lack of CD-8 responses may also be due to formulation and adjuvant choices as well. It would help to comment on the formulations assessed in reference 73.

We agree with the reviewer and added more information on vaccine type and formulation.

Figure 2 – This is a very good representation of the formulations.

We thank the reviewer for this positive feedback.

Line 514 – “strategies have”

This was changed accordingly.

Line 532 – “administer”

This was changed accordingly.

Line 540 – Suggested:

All in all, an improved understanding of the concept of heterologous prime-boost vaccinations would allow full assessment of potential suitability for use in a (pre-)pandemic context.

We thank the reviewer for this suggestion. Based on it, we partially changed the sentence to: All in all, an improved understanding of the concept of heterologous prime-boost vaccinations would allow to fully assess its potential suitability for use in a (pre-)pandemic context.

Figures 3 and 4 - The description and discussion of these figures could be improved and summarized more clearly. It was difficult to follow in the text. The figure legends are clear, however.

We thank the reviewer for this comment. Based on it, we added the information on what is plotted on which axis but we think describing the figures in more detail in the text would be redundant.

Reviewer 4 Report

It is important to know whether vaccines containing one influenza strain can provide cross-protectivity against antigenically distinct viruses. The MS ID vaccines-1463426 is a comprehensive well wrote and well-illustrated review of a number of potentially pandemic H5 influenza vaccines and prime-boost vaccination strategies. The heterologous vaccination part is especially of interest. The tables and figures are excellent and contain a lot of detailed information. I have just a few minor comments.

Point 1: Line 38. The authors referred to the paper by Taubenberger and Morens (2009) and stated that "Since 1918, four influenza pandemics occurred." What about the so-called "Russian flu" H1N1 pandemic (1977-1978)? Taubenberger and Morens (2009) did not mention it but other researchers, for instance, Enserink (2006) did (10.1126/science.312.5781.1725).

Point 2: Supplementary Figures 1 (Line 179) and 2 (Line 311) are available for download. However, I was not able to find Supplementary Tables 1 and 2 (Line 104-105). Were they attached during the submission process?

Point 3: Tables 1-2. I would recommend adding ClinicalTrials.gov Identifiers if available.

Point 4: From the column heading “Heterologous strains (clade)” of Table 1 it is not clear that it means heterologous antigen used for measuring antibodies in sera obtained after vaccination with “Vaccine (clade).”

Point 5: From the column heading “Heterologous strains (clade)” of Table 2 it is not clear that it means antigen used for measuring antibodies in sera obtained after revaccination.

Point 6: Figures 3-4. I would recommend indicating in the caption to the figures the numbers of references on papers based on which the graphs are built.

Author Response

It is important to know whether vaccines containing one influenza strain can provide cross-protectivity against antigenically distinct viruses. The MS ID vaccines-1463426 is a comprehensive well wrote and well-illustrated review of a number of potentially pandemic H5 influenza vaccines and prime-boost vaccination strategies. The heterologous vaccination part is especially of interest. The tables and figures are excellent and contain a lot of detailed information. I have just a few minor comments.

Point 1: Line 38. The authors referred to the paper by Taubenberger and Morens (2009) and stated that "Since 1918, four influenza pandemics occurred." What about the so-called "Russian flu" H1N1 pandemic (1977-1978)? Taubenberger and Morens (2009) did not mention it but other researchers, for instance, Enserink (2006) did (10.1126/science.312.5781.1725).

Taubenberger & Morens (2009) actually describe the Russian flu outbreak in their article but conclude that the current evidence suggests this was most likely not a natural pandemic, which is in line with the general consensus in the influenza field. However, because of the reviewer’s comment we revisited the article by Taubenberger & Morens (2009) and replaced the reference in line 38 with a more recent one, in which the Russian outbreak was also not considered to be a pandemic.

Point 2: Supplementary Figures 1 (Line 179) and 2 (Line 311) are available for download. However, I was not able to find Supplementary Tables 1 and 2 (Line 104-105). Were they attached during the submission process?

We thank the reviewer for this comment and apologise to have forgotten to upload the supplementary figures. The supplementary tables 1 and 2 are now incorporated with the revised version of the manuscript.

Point 3: Tables 1-2. I would recommend adding ClinicalTrials.gov Identifiers if available.

We thank the reviewer for this comment. The clinical trial.gov identifiers are now included in the supplementary tables 1 and 2.

Point 4: From the column heading “Heterologous strains (clade)” of Table 1 it is not clear that it means heterologous antigen used for measuring antibodies in sera obtained after vaccination with “Vaccine (clade).”

This was clarified as follows: Heterologous strains (clade) used for immunological endpoint(s).

Point 5: From the column heading “Heterologous strains (clade)” of Table 2 it is not clear that it means antigen used for measuring antibodies in sera obtained after revaccination.

This was clarified as above.  

Point 6: Figures 3-4. I would recommend indicating in the caption to the figures the numbers of references on papers based on which the graphs are built.

We thank the reviewer for this comment. However, since each panel is comprised of data of a multitude of studies, the caption would be far too extensive to be informative in our opinion. The information on each of references is already included in Supplementary figure 1 and 2, where the reference for each particular data point can be seen by hovering over the point.  
